# Factors associated with modern contraceptives uptake during the first year after birth in Ethiopia: A systematic review and meta-analysis

Gebi Husein Jima[1,2]*, Muhammedawel Kaso Kaso[1], R. G. Biesma-Blanco[2], Tegbar Yigzaw Sendekie[3], J. Stekelenburg[2,4]

1 Department of Public Health, College of Health Science, Arsi University, Asella, Ethiopia, 2 Department of Health Sciences, Global health, University of Groningen/University Medical Centre Groningen, Groningen, The Netherlands, 3 Jhpiego, Addis Ababa, Ethiopia, 4 Department of Obstetrics and Gynaecology, Medical Centre Leeuwarden, Leeuwarden, The Netherlands

* gebihussein@gmail.com

**Data Availability Statement:** All relevant data are within the manuscript and its Supporting information files.

## Abstract

Though postpartum family planning helps women to achieve the recommended birth interval before next pregnancy, its utilization in Ethiopia is low. Understanding drivers and barriers is key to improve postpartum family planning uptake. The aim of this systematic review and meta-analysis is to analyze and summarize predictors of postpartum family planning uptake, during the first year after birth, in Ethiopia. We conducted a systematic review and meta-analysis of observational studies published in English before April 16, 2021. We searched electronic sources like PubMed, MEDLINE, CINHAL Embase, Google and supplemented it with manual search. Two reviewers appraised independently the studies using the Joanna Briggs Institute Quality Assessment Tool for the observational studies. Data synthesis and analysis were conducted using Review Manager Version 5.3. The Cochrane Q test statistic and $I^2$ tests were used to assess the heterogeneity among the included studies. A random-effects and fixed effect model were used to calculate pooled Odds Ratio and its 95% CI. A total of 22 studies were included in the review. Better educational status of women[OR = 2.60; 95% CI: 2.15, 3.14], women's marital status [OR = 4.70; 95% CI: 1.51, 14.60], resumption of sexual intercourse [OR = 6.22; 95% CI: 3.01, 12.86], menses return [OR = 3.72; 95% CI: 1.98, 6.99], PPFP discussion with partner [OR = 2.53; 95% CI: 2.00, 3.20], women's previous PPFP information [OR = 4.93; 95% CI: 2.26, 10.76], PPFP counseling during ANC [OR = 3.95; 95% CI: 2.50, 6.23], having PNC [OR = 4.22; 95% CI: 2.80, 6.34], having experience of modern contraceptive use [OR = 2.90; 95% CI: 1.62, 5.19], facility birth [OR = 6.70; 95% CI: 3.15, 14.25], and longer interval after last delivery [OR = 0.37; 95% CI: 0.32, 0.43] were significantly associated with modern contraceptive uptake during postpartum period. Our systematic review identified modifiable factors and estimated their association with PPFP uptake. Since most of these factors are related to reproductive health characteristics and MNCH services, integrating PPFP into MNCH services particularly at primary health care unit may improve contraceptive uptake during postpartum period.

**Funding:** The author(s) received no specific funding for this work.

**Competing interests:** The authors have declared that no competing interests exist.

**Abbreviations:** ANC, Antenatal Care; aOR, Adjusted Odds Ratio; AOR, Adjusted Odds Ratio; CI, Confidence Interval; CPR, Contraceptives Prevalence Rate; DHS, Demographic Health Survey; EDHS, Ethiopian Demographic Health Survey; FP, Family Planning; MOH, Ministry of Health; OR, Odds Ratio; PNC, Postnatal Care; PPFP, Postpartum Family Planning; PRISMA, Preferred Reporting Items for Systematic Reviews and Meta-analysis; RMNCH, Reproductive, maternal, new-born and child health; SPNN, South Peoples, Nations and Nationalities.

**Systematic review registration**: PROSPERO: 2020: CRD42020159470.

## Introduction

The World Health Organization (WHO) defined postpartum family planning as "the prevention of unintended and closely spaced pregnancies through the first 12 months following childbirth" [1]. Globally, family planning is recognized as a key life-saving intervention for mothers and their children [1]. Inter-pregnancy intervals of at least 24 months (or birth intervals of nearly three years) are recommended due to the association between shorter intervals and increased risks for adverse outcomes to both mother and baby [2, 3]. Closely spaced pregnancies within the first year postpartum are the riskiest for mother and baby, resulting in increased risks for adverse outcomes, such as preterm, low birth weight and small-for-gestational age and malnutrition and stunting. Risk of child mortality is highest for very short birth-to-pregnancy intervals (<12 months) [1]. Family planning can avert more than 30% of maternal deaths and 10% of child mortality if couples space their pregnancies more than 2 years apart. If all couples waited 24 months to conceive again, under-five mortality would decrease by 13%. If couples waited 36 months, the decrease would be 25% [1].

Postpartum family planning has an important role to play in strategies to reduce the unmet need for family planning. Postpartum women are among those with the greatest unmet need for family planning. Yet they often do not receive the services they need to support longer birth intervals or reduce unintended pregnancy and its consequences [1]. According to an analysis of DHS data from 27 countries including Ethiopia, 95% of women who are 0–12 months postpartum want to avoid a pregnancy in the next 24 months but only 30% of them are using contraception [1].

Postpartum women are more likely to use the lactational amenorrhea method and injectables [4]. Exclusive or predominant breastfeeding offers protection against rapid fertility return during the first 6 months after child birth. But rates of exclusive breastfeeding drop off sharply in Ethiopia. EDHS 2012 still reported 74% of newborns (0–1 month) being exclusively breastfed, while 36% of infants 4–5 months old are exclusively breastfed [5]. However EDHS 2019 reported that only 59% of infants under 6 months were exclusively breastfed. The percentage of exclusively breastfed decreased sharply with age from 73% of infants age 0–1 months to 68% of those age 2–3 months and further, to 40% of infants age 4–5 months [6].

Like many low-income countries, the use of modern contraception has substantially increased in Ethiopia over the past two decades; from 6% of married women using modern contraception in 2000 to 41% in 2019 [6]. EDHS 2019 also reported a slight increase over 14 years in the use of any modern contraceptive method; from 14% in 2005 to 41% in 2019 [6]. Still, there is a sizable gap between the number of women desiring to prevent or delay pregnancy and the number using modern contraception, particularly among postpartum women, for whom that gap is 74% [5].

Ethiopia has one of the highest numbers of postpartum women not using modern contraception [7]. As a result, 22% of non-first births are less than 24 months after the previous birth and an additional 32% are 24–35 months after the previous birth [7–9]. A number of determinants have been reported for modern contraceptive uptake during postpartum period. In a report using reproductive calendar data from 43 Demographic and Health Surveys (DHS) representing 61 percent of the developing world's population reported a strong correlation between use of maternal health care and uptake of postpartum family planning [4].

Increasing use of postpartum family planning (PPFP) can reduce the number of short-interval pregnancies and associated risks [5]. Although PPFP is part of the Federal MOH costed implementation plan to increase contraceptive use, the practice gap suggests all levels of the health system need to prioritize meeting the contraceptive needs of postpartum women. Offering contraception to women immediately after birth in a facility is an important strategy [9, 10]. Yet in Ethiopia, the majority of women deliver at home, thus many postpartum women will not benefit from the integration of postpartum family planning services within facility-based childbirth services. Community-based family planning, which can be effective in improving contraceptive use generally, may be important for improving postpartum family planning services [10].

Understanding the drivers and barriers to PPFP uptake is key to improve use of modern contraceptives during the postpartum period. Various studies tried to determine the level of post-partum family planning (PPFP) utilization and associated factors in Ethiopia. However, there is huge variability in the findings. Moreover, although there were previous attempts to do a systematic review and meta-analysis, they included few studies and largely focused on one region of Ethiopia, limiting their generalizability to the country as a whole [11, 12]. In our review we tried to include primary studies conducted from all major regions in Ethiopia to address this limitation. Therefore, the aim of this systematic review and meta-analysis is to analyze and summarize predictors of postpartum family planning uptake in Ethiopia. The evidence generated from this study will help the Federal Ministry of Health, regional health bureaus and other stakeholders to improve modern contraceptive use during postpartum in Ethiopia.

## Materials and methods

### Protocol and registration

This systematic review and meta-analysis is conducted in accordance with the PRISMA 2020 statement: an updated guideline for reporting systematic reviews [13]. The review has been registered by the International prospective register of systematic reviews and the protocol is available in the Prospero database: (PROSPERO: 2020: CRD42020159470) (data in S1 File).

### Information source sand search strategy

The investigators searched electronic and non-electronic database sources. Published articles were retrieved from like PubMed/MEDLINE, CINHAL and Embase. We also directly searched on Google Scholar and Google. The search strategy included the use of Title/Abstract related to "postpartum family planning" or "PPFP" or "family planning service" "AND" "Determinants" or "Predicators" or "Factors associated" "AND" "Ethiopia". In addition, the investigators searched manually for grey literature and other relevant data sources such as email and unpublished theses/papers within the planned dates of coverage (S1 Table).

### Eligibility criteria

This search included all cross-sectional, case-control, and cohort studies on factors affecting postpartum family planning utilization in Ethiopia. The search covered studies conducted before April 16, 2021, conducted on humans, and written in English language.

### Study screening and selection

All searched articles were exported to the EndNote X8 citation manager. Duplicates were excluded. To identify eligible studies, titles and abstracts were screened. Two assessors (GH

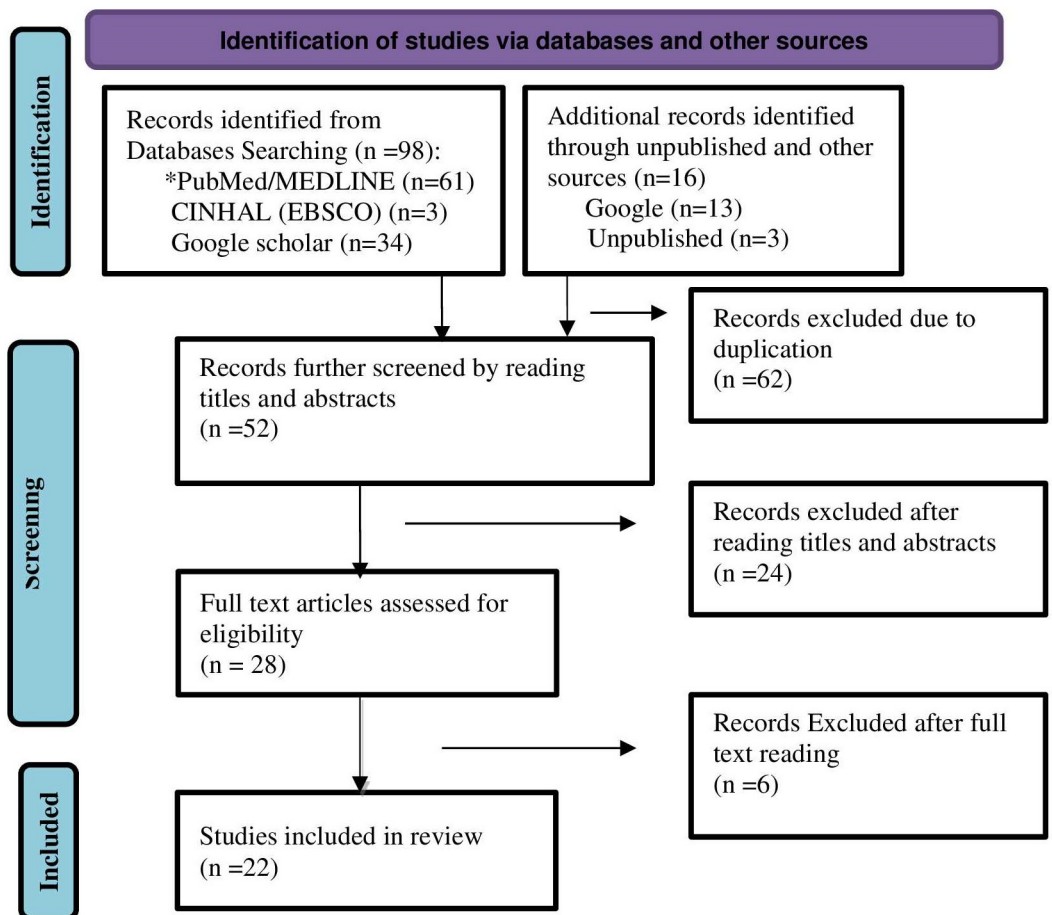

**Fig 1. Description of schematic presentation of the PRISMA flow diagram, 2021.** *During data base searching, automation tools were used; 23 records were excluded for not being human subject studies, 5 records were removed because full text was not available, and 1 record was removed because it was not a journal articles (SI).

and MKA) screened and evaluated studies independently. The titles and abstracts of studies that clearly mentioned the outcomes of the review (factors associated/determinants/predictors/ of postpartum family planning methods utilization) were considered for further evaluation to be included in the study. Moreover, the full-texts of the selected studies were further evaluated based on objectives, methods, participants/population and key findings which are factors associated/affecting/determinants/predicators/ of postpartum family planning methods utilization. The two assessors (GH and MKA) independently evaluated the quality of the included studies with the JBI Critical Appraisal Checklist for prevalence studies, case control Studies and cohort studies [14, 15]. The overall study selection process is presented in the PRISMA statement flow diagram [13] (Fig 1).

## Data collection process

From the selected 22 studies, data were carefully extracted by two independent investigators (GH and MKA) using a data extraction templates, prepared in Microsoft word 2010 for this purpose. Information about the included studies, like author, year of publication, region, setting, design, sample size, and factors associated with postpartum family planning utilization

were retrieved. Authors double checked the accuracy of the data extraction by comparing the results with the data extraction summary (Table 1).

The quantitative data, such as total sample size (n), magnitude of contraceptive uptake during postpartum period with its 95% CI, and specific factors associated with contraceptive uptake during postpartum period (aOR and its 95% CI), were extracted from the included research articles and summarized using Microsoft Excel 2010 for meta-analysis and synthesis (S2 Table). In addition, frequencies of each category of the considered variables were organized in Microsoft word 2010 template prepared for this purpose (S3 Table).

## Outcomes of the study

The outcomes of interest of this systematic review and meta-analysis are factors affecting modern contraceptives uptake during the first year after birth.

## Risk of bias in individual studies

Investigators critically evaluated the risk of bias in individual studies using the Joanna Briggs Institute Quality Assessment Tool for observational studies [14, 15]. To minimize the risk of bias, comprehensive searches (electronic/database search and manual search) were conducted which included published, unpublished, facility-based, and community-based studies and theses. Cooperative work of the authors was also critical for reducing bias. These cooperatives included setting a schedule for the selection of articles based on the clear objectives and eligibility criteria, deciding the quality of the article, regularly evaluating the review process, and extracting and compiling the data. Publication bias was explored using visual inspection of the funnel plot. Besides, Egger's Regression Test was carried out to check statistically symmetry of the funnel plot [37].

## Synthesis of data

Data synthesis and statistical analysis were conducted using Review Manager (RevMan) version 5.3.5. A meta-analysis of observational studies was carried out, based on the recommendations of the $I^2$ statistic described by Higgins et al. An $I^2$ of 75/100% and above was considered as considerable heterogeneity among the included studies [38]. Based on the value of heterogeneity test, either a random or fixed effect model was used to decide whether there was a statistically significant association between modern contraceptive uptake during the postpartum period and the specific factors considered. The results of the review are reported according to the PRISMA guidelines. The findings of the included studies were presented using a narrative synthesis and meta-analysis chart.

# Results

## Description of review studies

A total of 114 articles were identified through electronic and manual searches. From these records, 22 articles fulfilled the eligibility criteria and were included in the systematic review and meta-analysis (Table 1).

## Factors associated with modern contraceptives uptake during postpartum period

From the selected 22 studies, we retrieved factors associated with modern contraceptive uptake during postpartum period: women's education status, women's marital status, resumption of sexual activity, return of menses after birth, discussion about family planning with a partner,

**Table 1. Description of studies included in the systematic review and meta-analysis.**

| No. | Authors | Year | Region | Setting of the study | Design of the study | Sample size | Specific factors associated with postpartum family planning utilization |
|---|---|---|---|---|---|---|---|
| 1. | Gejo NG et al, 2019 [16] | 2018 | SNNP | Facility-based | Cross-sectional | 368 | Educational status of mothers, resumption of sex, menses resumption & duration postpartum period |
| 2 | Ashebir W et al, 2020 [17] | 2017 | Amhara | Community-based | Cross-sectional | 686 | women's level of education, discussing FP methods with partner, knowing menses return after birth, ever heard about modern FP methods, contacting health professionals |
| 3 | Abraha TH et al,2017 [18] | 2015 | Tigray | Community-based | Cross-sectional | 601 | secondary and tertiary education levels, family planning counseling during prenatal and postnatal care, having postnatal care), resuming sexual activity, menses returning after birth, experiencing problems with previous contraceptive |
| 4 | Dagnew GW et al,2020 [19] | 2016 | Ethiopia | Community-based | Cross-sectional | 2304 | urban residents, secondary or higher education, women who attended 1–3, 4 or more ANC visits, women who delivered at a health facility, the last child was no more wanted, women who decided for contraceptive use, recent child was male |
| 5 | Taye EB et al, 2018 [20] | 2018 | Amhara | Community-based | Cross-sectional | 550 | age of the mother (25–29), married women, return of menses &previous history of family planning |
| 6 | Dona A et al, 2018 [21] | 2017 | SNNP | Community-based | Cross-sectional | 695 | Antenatal care, postnatal care, communication on contraceptive methods & resumption of menses after delivery |
| 7 | Berta M et al, 2018 [22] | 2015 | Amhara | Facility-based | Cross-sectional | 404 | Menstruating, resumption of sex, 37–51 weeks of postpartum period, husband approval of contraceptive & current knowledge on FP |
| 8 | Teka TT et al, 2018 [23] | 2015 | Oromia | Community-based | Cross-sectional | 616 | Women who had four and more antenatal care visits, received post-natal care & desiring less number of children |
| 9 | Mengesha ZB et al, 2015 [24] | 2012 | Amhara | Community-based | Cross-sectional | 816 | Women who delivered with the assistance of a skilled attendant, attended postnatal care services, & Secondary and above level of the husband's education |
| 10 | Abera Y et al, 2015 [25] | 2013 | Amhara | Community-based | Cross-sectional | 703 | Resumption of menses, age ≤24 years, having antenatal care, & duration of 7–9 months after delivery |
| 11 | Emiru AA et al, 2020 [26] | 2018 | Amhara | Community-based | Cross-sectional | 1281 | Women's education, four or more antenatal care, early initiation of antenatal care,& early postnatal checkup |
| 12 | Demie TG et al, 2018 [27] | 2016 | Amhara | Facility-based | Cross-sectional | 248 | Resumption of sexual intercourse, resuming of sex before six month, & return of menses |
| 13 | Gebremedhin, AY et al, 2018 [28] | 2015 | Addis Ababa | Community-based | Cross-sectional | 803 | Marriage, menses resumption after birth, length of time after delivery, & history of contraceptive use before last pregnancy |
| 14 | Tafa L et al, 2018 [29] | 2018 | Addis Ababa | Facility-based | Cross-sectional | 625 | Previous family planning (FP) information, FP information from health facility visit, antenatal care, counseling on FP at postnatal care, menses resumption after birth, & commencing sexual activity after birth |
| 15 | Wassihun, B et al, 2021 [30] | 2019 | SNNP | Facility-based | Cross-sectional | 408 | antenatal care visit, planned pregnancy, married, & college and above level educational status |
| 16 | Tafere TE et al, 2018 [31] | 2015–2016 | Amhara | Facility-based | prospective follow up study | 970 | Counseling on PPFP during antenatal care, birth preparedness and complication readiness plan, counseling on breast feeding at least once during ANC visits, satisfaction with antenatal care service, & post-natal care visit |
| 17 | Seifu B et al, 2020 [32] | 2019 | Oromia | Community-based | Cross-sectional | 367 | Higher education level, history of family planning utilization, having ANC follow up, knowledge of PPFP, grand multipara women, & received routine PNC service |
| 18 | Mihretie GN et al, 2020 [33] | 2019 | Amhara | Community-based | Cross-sectional | 402 | Maternal educational status, menses return, less than four alive children, postnatal care follow-up, length of time after delivery, & PPFP knowledge |
| 19 | Jima GH et al, 2020 [34] | 2018 | Oromia | Community-based | Cross-sectional | 1162 | partners completed secondary education, FP counseling during ANC visit, & No FP counseling at PNC services |
| 20 | Kenea LB et al, 2021 [35] | 2019 | Oromia | Community-based | Cross-sectional | 597 | Mothers with four and above Antenatal care(ANC) visits & Mothers who were delivered in hospital |

*(Continued)*

**Table 1.** (Continued)

| No. | Authors | Year | Region | Setting of the study | Design of the study | Sample size | Specific factors associated with postpartum family planning utilization |
|---|---|---|---|---|---|---|---|
| 21 | Tefera K et al, 2020 [36] | 2018 | SNNP | Community-based | Cross-sectional | 381 | received FP education after delivery at immunization service, received FP education after delivery at post natal ward, mothers' discussion with husband on family planning issue, Experienced side effects while using FP, 0–6 month of postnatal period, 1–4 number of pregnancy, &History of abortion |
| 22 | Belete GA et al, 2019[Unpublished] | 2019 | Amhara | Community-based | Cross-sectional | 400 | Secondary school educational level, higher educational level, previous history of abortion, having three and four antenatal care visit, family planning counseling during antenatal care, having postnatal care, menses returning after birth, resuming sexual activity |

women have ever heard about PPFP, women were counseled about PPFP during ANC, ANC visits, skilled delivery, PNC visits, previous contraceptive use and timing after the last delivery.

The following factors appeared to be significantly associated with modern contraceptives uptake during the first year after birth, after random or fixed effect model based on the value of heterogeneity test, to be significantly associated with modern contraceptives uptake during postpartum period in Ethiopia: women's educational status, women's marital status, resumption of sex after birth, menses return after birth, PPFP discussion with partner, women ever heard about PPFP, PPFP counseling during ANC visit, PNC visits, skilled delivery, previous contraceptive use and timing after the last delivery. But having ANC follow-up did not show statistically significant association with modern contraceptives uptake during postpartum period. The details of these results are presented below.

**Women's educational level.** Women who had formal education were more than two times more likely to utilize modern contraceptives during postpartum period compared to women who did not have formal education [OR = 2.60; 95% CI: 2.15, 3.14, P < 0.00001]. The heterogeneity test showed an $I^2$ value of only 36%, and therefore the fixed effect model was considered for the analysis (Fig 2).

**Women's marital status.** The odds of using contraceptives during the postpartum period was about four times greater among married women compared to women who were not married [OR = 4.34; 95% CI: 2.05, 9.17, P = 0.0001]. The random effects model was used for the

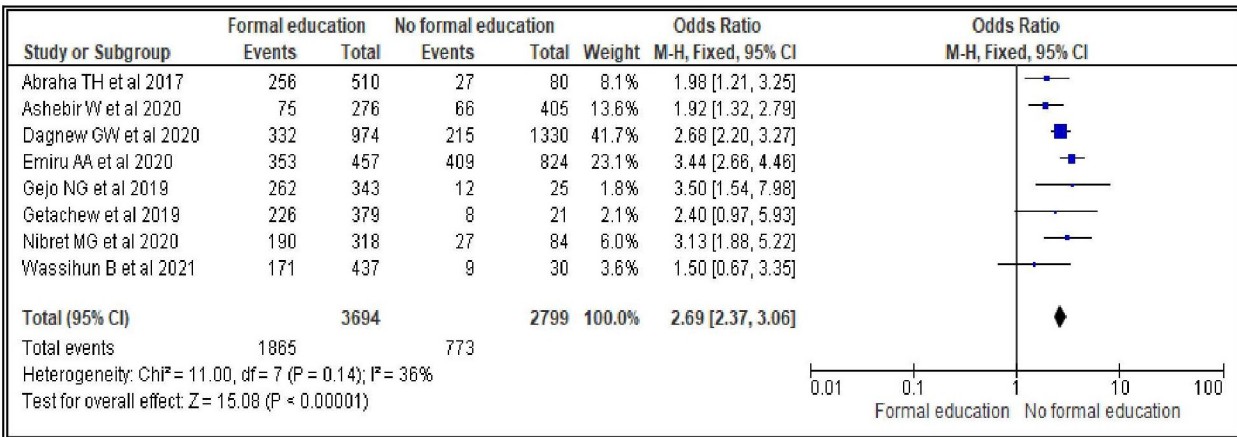

**Fig 2. Association between women's educational status and modern contraceptive uptake during postpartum period, 2021.**

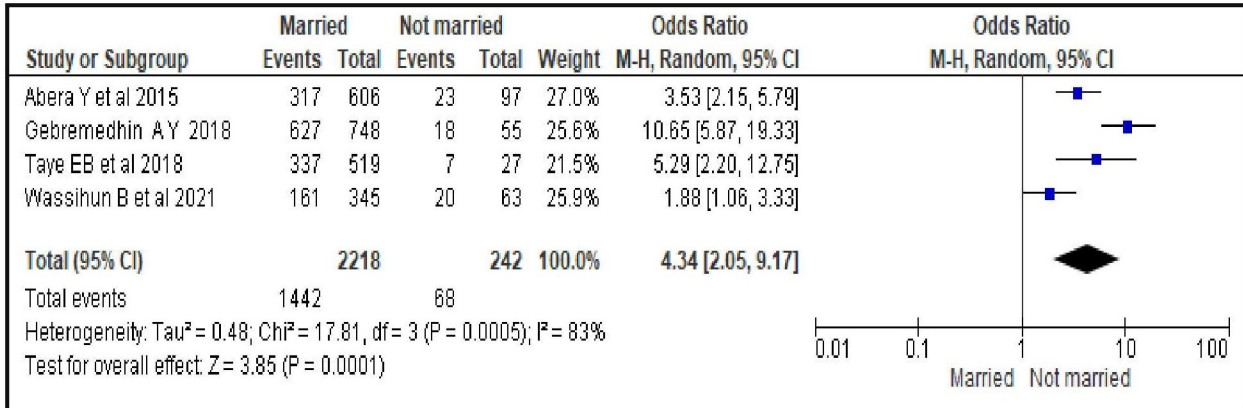

**Fig 3. Association between women's marital status and modern contraceptive uptake during postpartum period, 2021.**

analysis since the heterogeneity test showed an overall $I^2$ value of 83%. The 95% confidence interval was also found to be reasonably wide and therefore the random effect was used (Fig 3).

**Resumption of sexual intercourse after birth.** Women who resumed sexual intercourse after birth were six times more likely to initiate modern contraceptive during postpartum period compared to those who did not resume [OR = 6.22; 95% CI: 3.01, 12.86, P < 0.00001]. The random effects model was used since the heterogeneity test between the considered studies was found to be significant ($I^2$ = 93%, p <0.0001). In addition, the 95% confidence interval of the summary effect was wider, which is another reason why random effect was used (Fig 4).

**Menses return after birth.** The odds of initiating contraceptive use during the postpartum period was greater by nearly four-folds among women whose menses returned after birth compared to their counterparts [OR = 3.72; 95% CI: 1.98, 6.99, P < 0.0001]. Random effect model was used as the heterogeneity test was significant ($I^2$ = 96%, p <0.0001) (Fig 5).

**Discussion about family planning with partner.** Women who discussed family planning with their partner were 2.5 times more likely to utilize modern contraceptives during postpartum period than women who did not discuss [OR = 2.53; 95% CI: 2.00, 3.20, P < 0.00001]. Fixed effect model was used for the analysis as there was no statistically significant heterogeneity among the included studies for this factor analysis ($I^2$ = 44%, P = 0.17) (Fig 6).

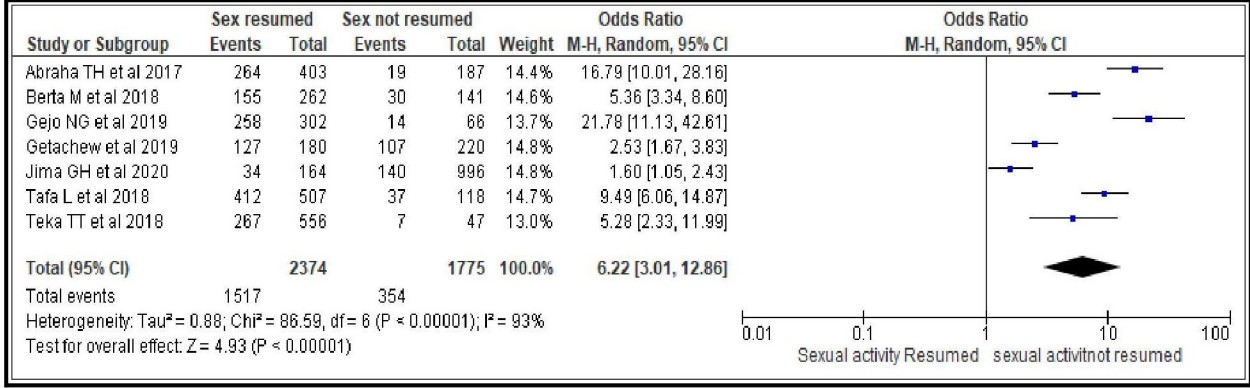

**Fig 4. Association between resumption of sexual activity after birth and modern contraceptive uptake during postpartum period, 2021.**

| Study or Subgroup | Menses returned | | Menses not returned | | Weight | Odds Ratio M-H, Random, 95% CI | Odds Ratio M-H, Random, 95% CI |
|---|---|---|---|---|---|---|---|
| | Events | Total | Events | Total | | | |
| AbAbraha TH 2017 | 156 | 205 | 127 | 385 | 9.1% | 6.47 [4.40, 9.50] | |
| Abera Y 2015 | 270 | 373 | 70 | 330 | 9.2% | 9.74 [6.87, 13.79] | |
| Ashebir W 2020 | 86 | 272 | 55 | 409 | 9.1% | 2.98 [2.03, 4.36] | |
| Berta M 2018 | 114 | 163 | 71 | 241 | 9.0% | 5.57 [3.61, 8.60] | |
| Dona A 2018 | 186 | 484 | 31 | 200 | 9.1% | 3.40 [2.23, 5.20] | |
| Gebremedhin A Y 2018 | 226 | 361 | 379 | 442 | 9.2% | 0.28 [0.20, 0.39] | |
| Gejo NG 2019 | 243 | 287 | 29 | 81 | 8.8% | 9.90 [5.68, 17.27] | |
| Getachew Andualem Belete 2019 | 158 | 219 | 76 | 181 | 9.1% | 3.58 [2.36, 5.43] | |
| Nibret Mihretie G 2020 | 127 | 167 | 90 | 235 | 9.0% | 5.12 [3.29, 7.96] | |
| Tafa L 2018 | 334 | 421 | 115 | 204 | 9.2% | 2.97 [2.07, 4.27] | |
| Taye EB 2018 | 229 | 300 | 115 | 249 | 9.2% | 3.76 [2.61, 5.41] | |
| | | | | | | | |
| Total (95% CI) | | 3252 | | 2957 | 100.0% | 3.72 [1.98, 6.99] | |
| Total events | 2129 | | 1158 | | | | |
| Heterogeneity: Tau² = 1.09; Chi² = 277.35, df = 10 (P < 0.00001); I² = 96% | | | | | | | |
| Test for overall effect: Z = 4.09 (P < 0.0001) | | | | | | | |

**Fig 5. Association between menses return after birth and modern contraceptive uptake during postpartum period, 2021.**

**Ever heard about modern contraceptives use during postpartum period.** Having ever heard of postpartum family planning increased the odds of a woman using modern contraceptive during postpartum period by about 5-fold [OR = 4.93; 95% CI: 2.26, 10.76, P < 0.0001]. Random effect model was used as the heterogeneity between the included studies was significant (I² = 84%, p <0.0001) (Fig 7).

**PPFP counseling during ANC follow-up.** Women who were counseled on PPFP during their ANC follow-up were about four times more likely to use modern contraceptives during postpartum period compared to those who were not counseled [OR = 3.95; 95% CI: 2.50, 6.23,

| Study or Subgroup | FP discussed | | FP not discussed | | Weight | Odds Ratio M-H, Fixed, 95% CI | Odds Ratio M-H, Fixed, 95% CI |
|---|---|---|---|---|---|---|---|
| | Events | Total | Events | Total | | | |
| Ashebir W et al 2020 | 80 | 267 | 61 | 414 | 36.5% | 2.48 [1.70, 3.61] | |
| Dona A et al 2018 | 152 | 396 | 65 | 288 | 50.5% | 2.14 [1.52, 3.01] | |
| Kebede T et al 2020 | 133 | 295 | 14 | 86 | 13.0% | 4.22 [2.28, 7.82] | |
| | | | | | | | |
| Total (95% CI) | | 958 | | 788 | 100.0% | 2.53 [2.00, 3.20] | |
| Total events | 365 | | 140 | | | | |
| Heterogeneity: Chi² = 3.59, df = 2 (P = 0.17); I² = 44% | | | | | | | |
| Test for overall effect: Z = 7.80 (P < 0.00001) | | | | | | | |

**Fig 6. Association between discussion about family planning with partner and modern contraceptive uptake during postpartum period, 2021.**

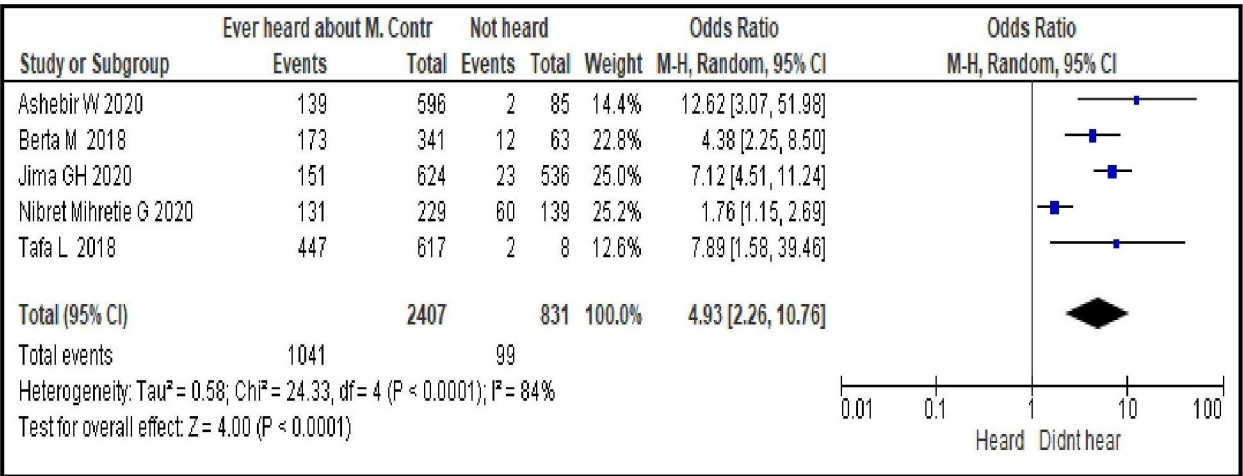

**Fig 7. Association between women's previous information about modern contraceptive use during postpartum period and its uptake, 2021.**

P < 0.00001]. Random effect model was used since the heterogeneity test between the considered studies was significant ($I^2$ = 86%, p <0.0001)(Fig 8).

**Having experience of modern contraceptive use before last pregnancy.** Women who had experience with usingmodern contraceptive before the last pregnancy were about three times more likely to use modern contraceptives during the postpartum period compared to who did not have an experience [OR = 2.90; 95% CI: 1.62, 5.19, P < 0.00001]. Random effect model was used as the heterogeneity test of the included studies in to the analysis was significant ($I^2$ = 89%, P = 0.0003) (Fig 9).

**Gave birth in a facility.** Women who gave birth in a health facility were nearly seven times more likely to use modern contraceptives during postpartum period than those who did not give birth in a facility [OR = 6.70; 95% CI: 3.15, 14.25, P < 0.00001]. Random effect model

| Study or Subgroup | Couseled Events | Total | Not counseled Events | Total | Weight | Odds Ratio M-H, Random, 95% CI |
|---|---|---|---|---|---|---|
| AbAbraha TH 2017 | 217 | 322 | 66 | 268 | 20.5% | 6.33 [4.40, 9.09] |
| Getachew Andualem Belete 2019 | 97 | 127 | 137 | 273 | 18.8% | 3.21 [2.00, 5.15] |
| Jima GH 2020 | 161 | 592 | 33 | 569 | 20.0% | 6.07 [4.09, 9.01] |
| Tafa L 2018 | 237 | 302 | 212 | 323 | 20.5% | 1.91 [1.33, 2.73] |
| Tafere T.E 2018 | 72 | 187 | 85 | 636 | 20.3% | 4.06 [2.80, 5.89] |
| Total (95% CI) | | 1530 | | 2069 | 100.0% | 3.95 [2.50, 6.23] |
| Total events | 784 | | 533 | | | |

Heterogeneity: Tau² = 0.23; Chi² = 27.64, df = 4 (P < 0.0001); I² = 86%
Test for overall effect: Z = 5.89 (P < 0.00001)

**Fig 8. Association between postpartum family planning (PPFP) counseling during ANC follow-up and modern contraceptive uptake during postpartum period, 2021.**

| Study or Subgroup | Had contr. experience | | Had no contr. experience | | | Odds Ratio | Odds Ratio |
| | Events | Total | Events | Total | Weight | M-H, Random, 95% CI | M-H, Random, 95% CI |
|---|---|---|---|---|---|---|---|
| Ashebir W 2020 | 113 | 485 | 28 | 196 | 19.9% | 1.82 [1.16, 2.86] | |
| Gebremedhin A Y 2018 | 557 | 629 | 88 | 174 | 20.6% | 7.56 [5.14, 11.12] | |
| Getachew Andualem Belete 2019 | 154 | 242 | 80 | 158 | 20.4% | 1.71 [1.14, 2.56] | |
| Seifu B 2020 | 121 | 249 | 23 | 105 | 19.1% | 3.37 [1.99, 5.70] | |
| Taye EB 2018 | 302 | 451 | 42 | 95 | 20.0% | 2.56 [1.63, 4.01] | |
| | | | | | | | |
| Total (95% CI) | | 2056 | | 728 | 100.0% | 2.90 [1.62, 5.19] | |
| Total events | 1247 | | 261 | | | | |
| Heterogeneity: Tau² = 0.39; Chi² = 34.95, df = 4 (P < 0.00001); I² = 89% | | | | | | | |
| Test for overall effect: Z = 3.59 (P = 0.0003) | | | | | | | |

0.01  0.1  1  10  100
Had experience   Had no experince

**Fig 9. Association between having experience of modern contraceptive use before and modern contraceptive uptake during postpartum period, 2021.**

was used as the heterogeneity test of the included studies was significant ($I^2$ = 93%, p <0.0001) (Fig 10).

**Having PNC follow-up.** The chances of using modern contraceptive during postpartum period were more than four times greater by women who had PNC follow-ups compared to those who didn't have a follow-ups [OR = 4.22; 95% CI: 2.80, 6.34, P < 0.0001]. Random effect model was used as the heterogeneity between the included studies was significant ($I^2$ = 90%, p <0.00001) (Fig 11).

**Length of time after last delivery.** Women who gave birth under 6 months ago were less likely to initiate modern contraceptive compared to those who gave birth 6 or more months ago [OR = 0.37; 95% CI: 0.32, 0.43, P < 0.00001]. Fixed effect model was used for the analysis as there was no statistically significant heterogeneity among the included studies for this factor analysis ($I^2$ = 30%, P = 0.21) (Fig 12).

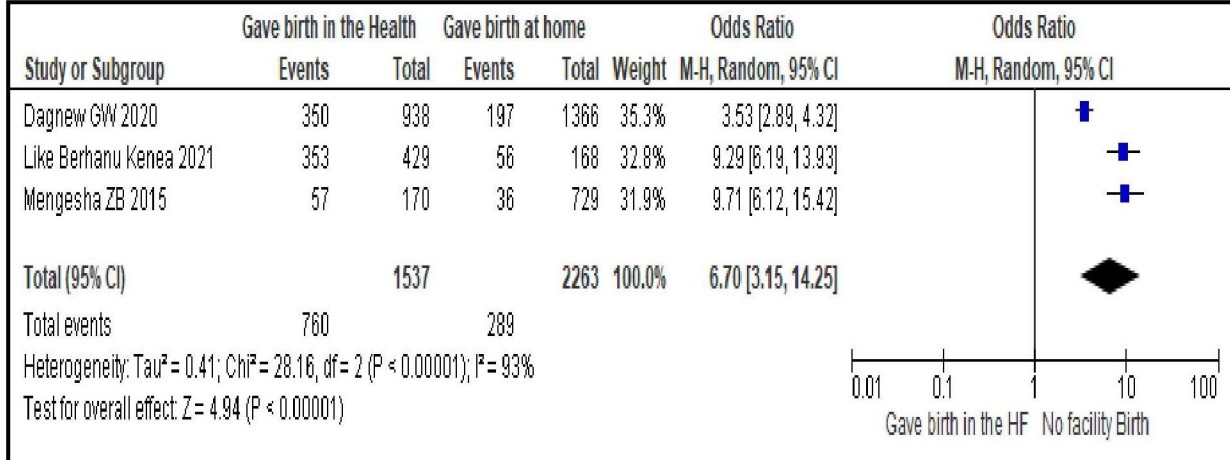

| Study or Subgroup | Gave birth in the Health | | Gave birth at home | | | Odds Ratio | Odds Ratio |
| | Events | Total | Events | Total | Weight | M-H, Random, 95% CI | M-H, Random, 95% CI |
|---|---|---|---|---|---|---|---|
| Dagnew GW 2020 | 350 | 938 | 197 | 1366 | 35.3% | 3.53 [2.89, 4.32] | |
| Like Berhanu Kenea 2021 | 353 | 429 | 56 | 168 | 32.8% | 9.29 [6.19, 13.93] | |
| Mengesha ZB 2015 | 57 | 170 | 36 | 729 | 31.9% | 9.71 [6.12, 15.42] | |
| | | | | | | | |
| Total (95% CI) | | 1537 | | 2263 | 100.0% | 6.70 [3.15, 14.25] | |
| Total events | 760 | | 289 | | | | |
| Heterogeneity: Tau² = 0.41; Chi² = 28.16, df = 2 (P < 0.00001); I² = 93% | | | | | | | |
| Test for overall effect: Z = 4.94 (P < 0.00001) | | | | | | | |

0.01  0.1  1  10  100
Gave birth in the HF   No facility Birth

**Fig 10. Association between giving birth in the facility and modern contraceptive uptake during postpartum period, 2021.**

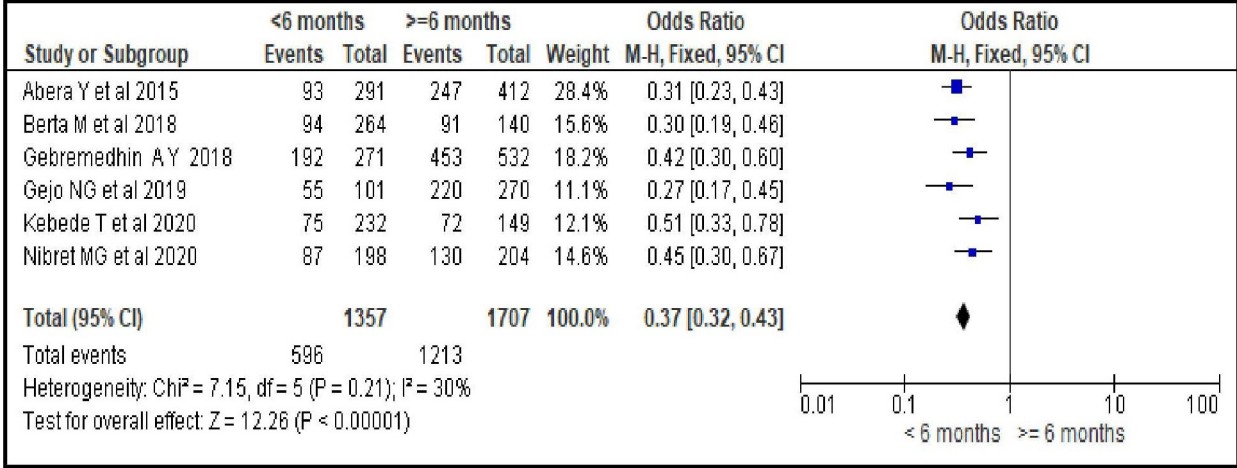

**Fig 11. Association between women's PNC follow-up during last pregnancy and modern contraceptive uptake during postpartum period, 2021.**

## Publication bias

In order to check publication bias among the included studies, funnel plot and Egger's test were carried out. The shape of the funnel plots was symmetrical indicating publication bias was not observed (Figs 13–23).

## Discussion

The aim of this systematic review and meta-analysis was to synthesize factors associated with modern contraceptive uptake during the 12 months of the postpartum period among

**Fig 12. Association between length of time after last delivery and modern contraceptive uptake during postpartum period, 2021.**

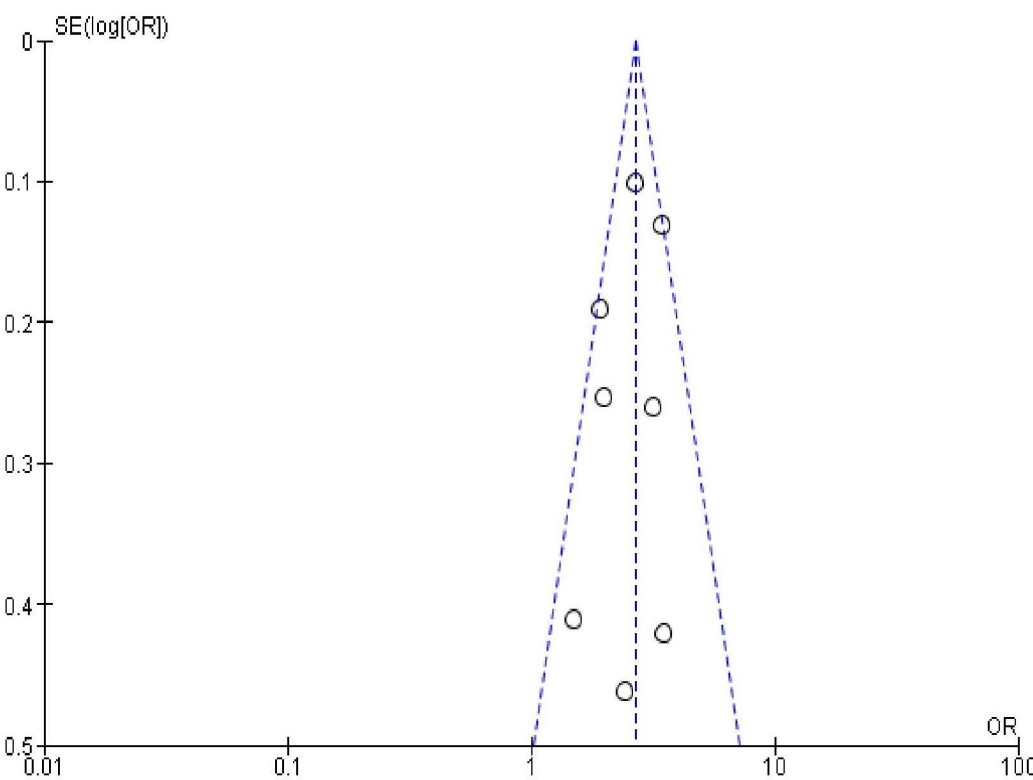

**Fig 13. Publication bias on women's educational status.**

Ethiopian women. A number of modifiable factors were identified and their association with modern contraceptive uptake in the first year after birth was estimated.

In our systematic review, most of the identified factors are related to sexual and reproductive health activities and services. Socio-demographic factors are also among the factors.

Reproductive health services related factors are PPFP counseling during ANC follow-up [OR = 3.95; 95% CI: 2.50, 6.23], having experience of modern contraceptive use before last pregnancy [OR = 2.90; 95% CI: 1.62, 5.19], giving birth in a facility [OR = 6.70; 95% CI: 3.15, 14.25], having PNC follow-up [OR = 4.22; 95% CI: 2.80, 6.34] and length of time after last delivery [OR = 0.37; 95% CI: 0.32, 0.43]. This group of factors showed more strong association with modern contraceptive uptake in the first year after birth compared to sexual and reproductive and socio-demographic characteristics as evidenced by the bigger magnitude of the OR and the narrow feature of its 95% CI. Particularly giving birth in a facility, having PNC follow-up and PPFP counseling during ANC follow-up are strongest predictors. These findings imply the importance of strengthening and integrating key RMNCH services so that a postpartum woman coming for a specific RMNCH service could be also counseled for contraceptives and given the method of their choice. In Ethiopia, these services are given separately in different rooms. A health care provider working in labor and delivery room provides only services related to labor and delivery and does not discuss about other RMNCH services, particularly contraceptives, with his/her client. This does not facilitate contraceptive uptake. Integrating family planning and RMNCH services is an ideal strategy for women who are at risk of a subsequent pregnancy and motivated to prevent another immediate pregnancy. This integration increases women's use of contraception in different countries, which leads to better birth spacing and improves the health of women and their infants. It is also cost-effective, can save the

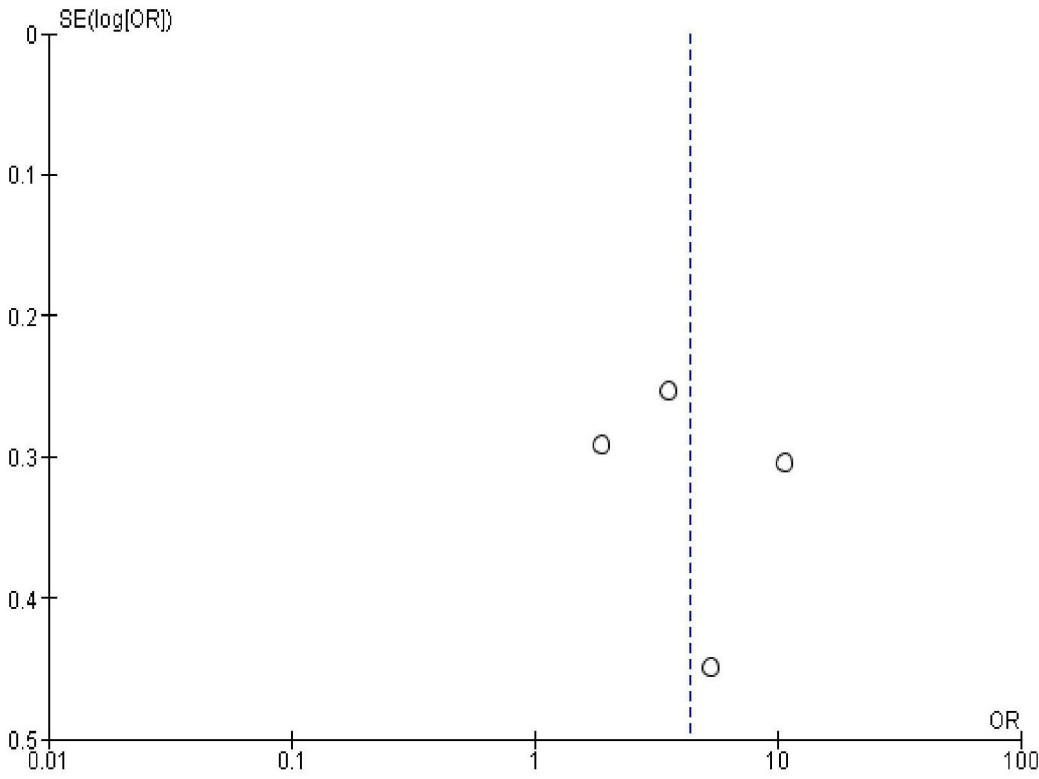

**Fig 14. Publication bias on marital status.**

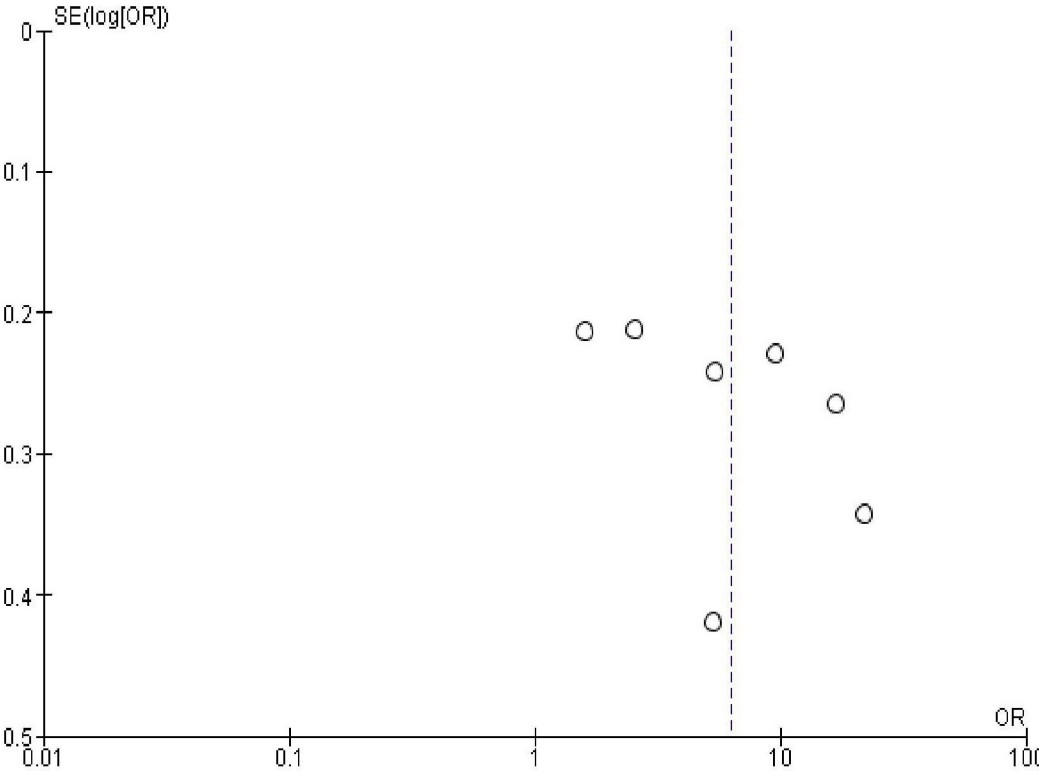

**Fig 15. Publication bias on resumption of sexual activity after the last birth.**

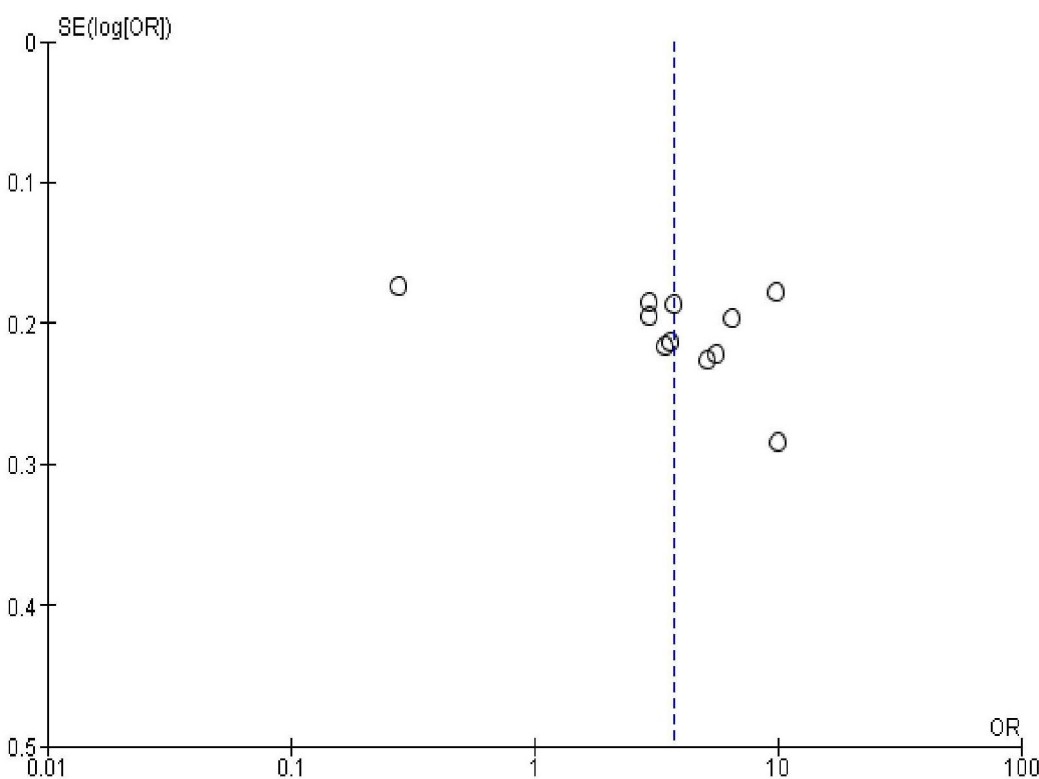

**Fig 16. Publication bias on menses return after birth.**

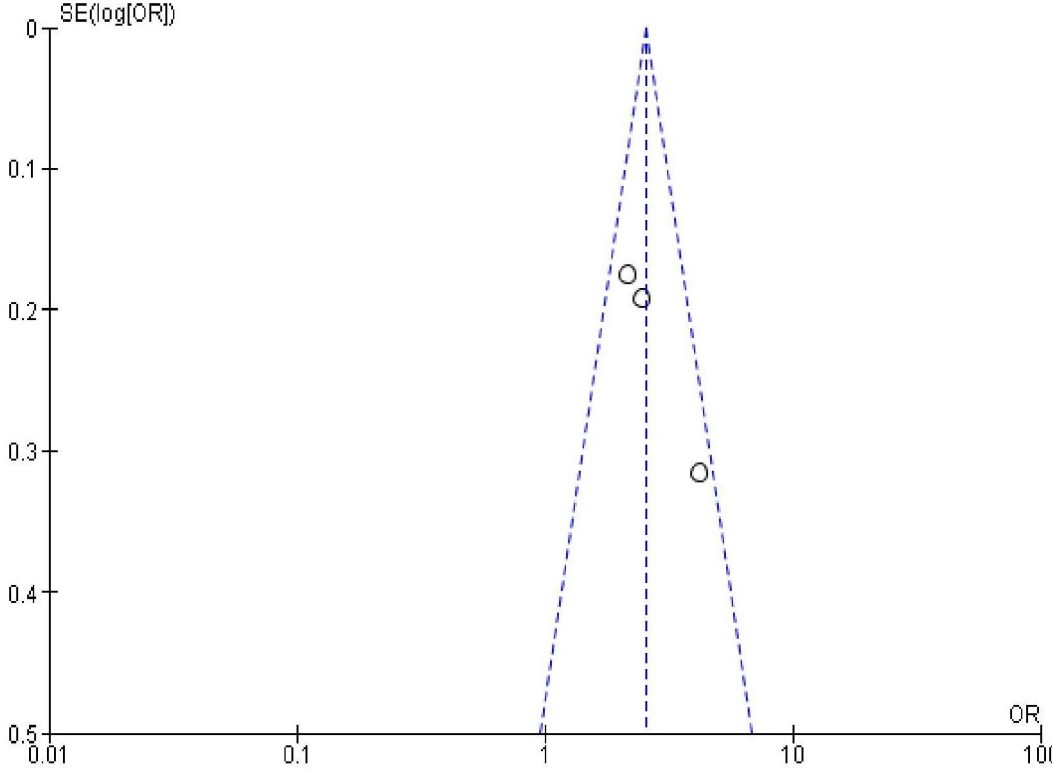

**Fig 17. Publication bias on discussing FP methods with partner.**

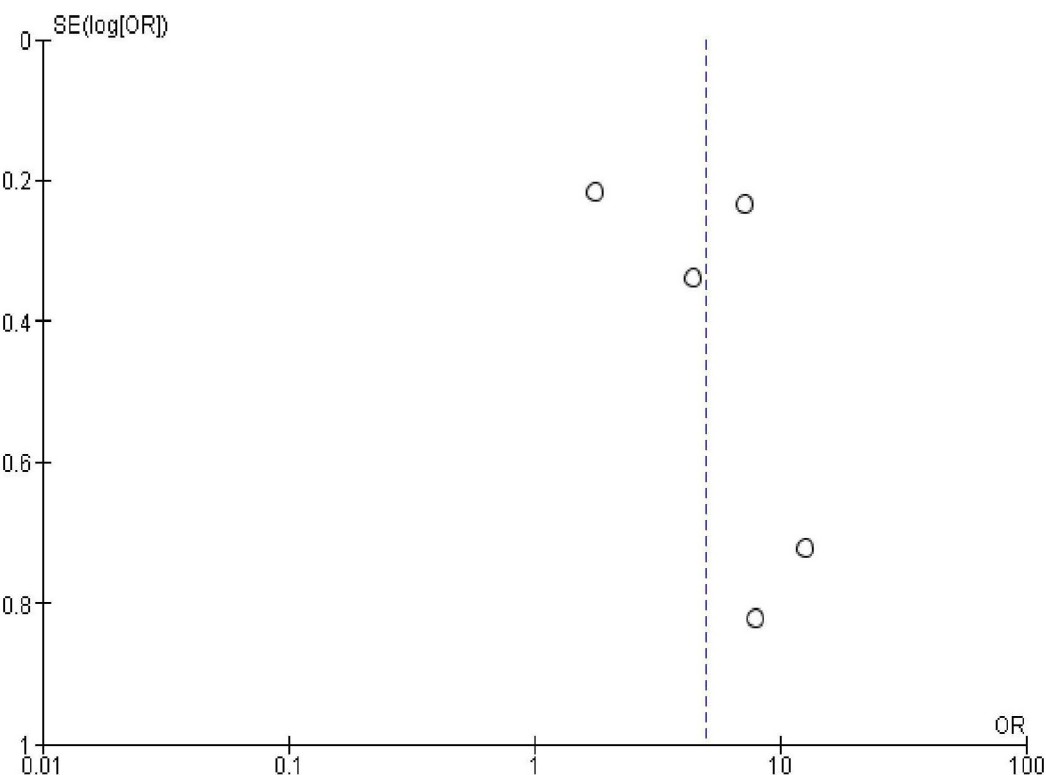

**Fig 18. Publication bias on ever heard about modern FP methods.**

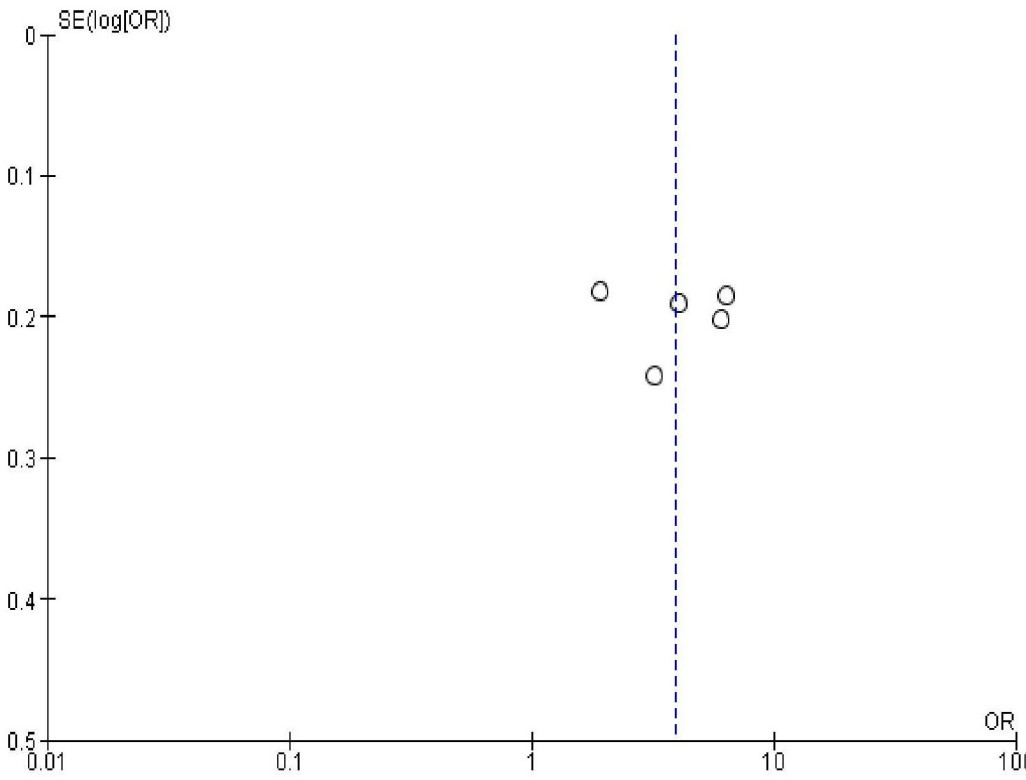

**Fig 19. Publication bias on PPFP counseling during ANC visit.**

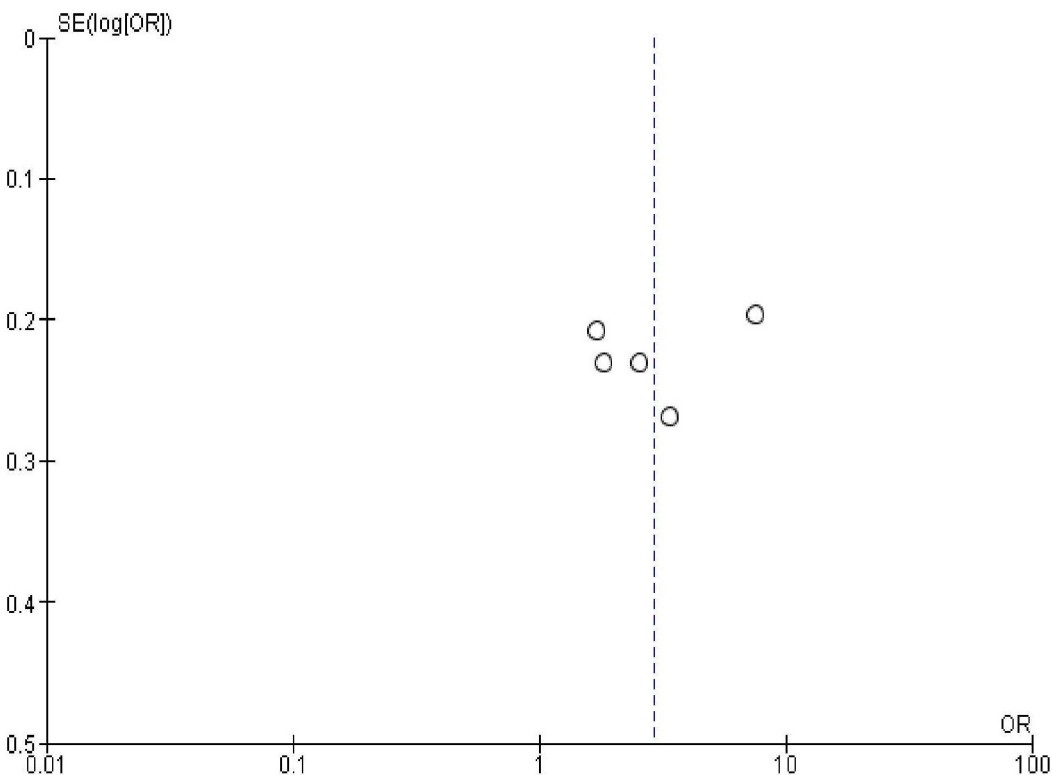

**Fig 20. Publication bias on having experiencing of contraceptive use before.**

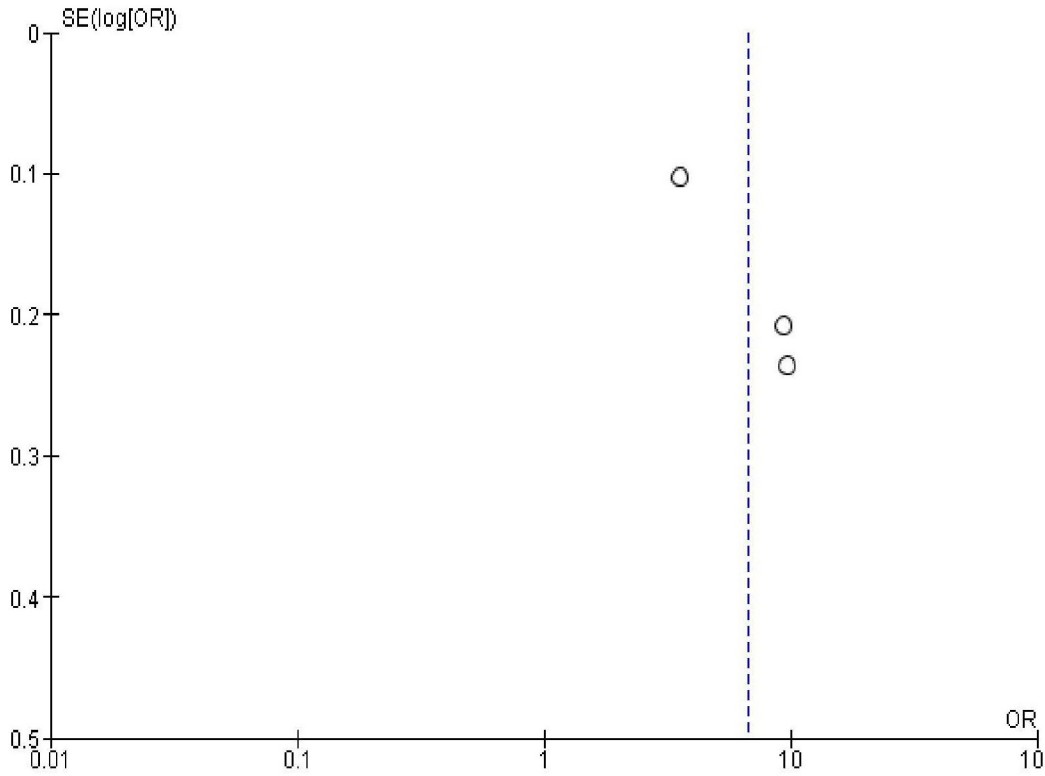

**Fig 21. Publication bias on birth in the health facility.**

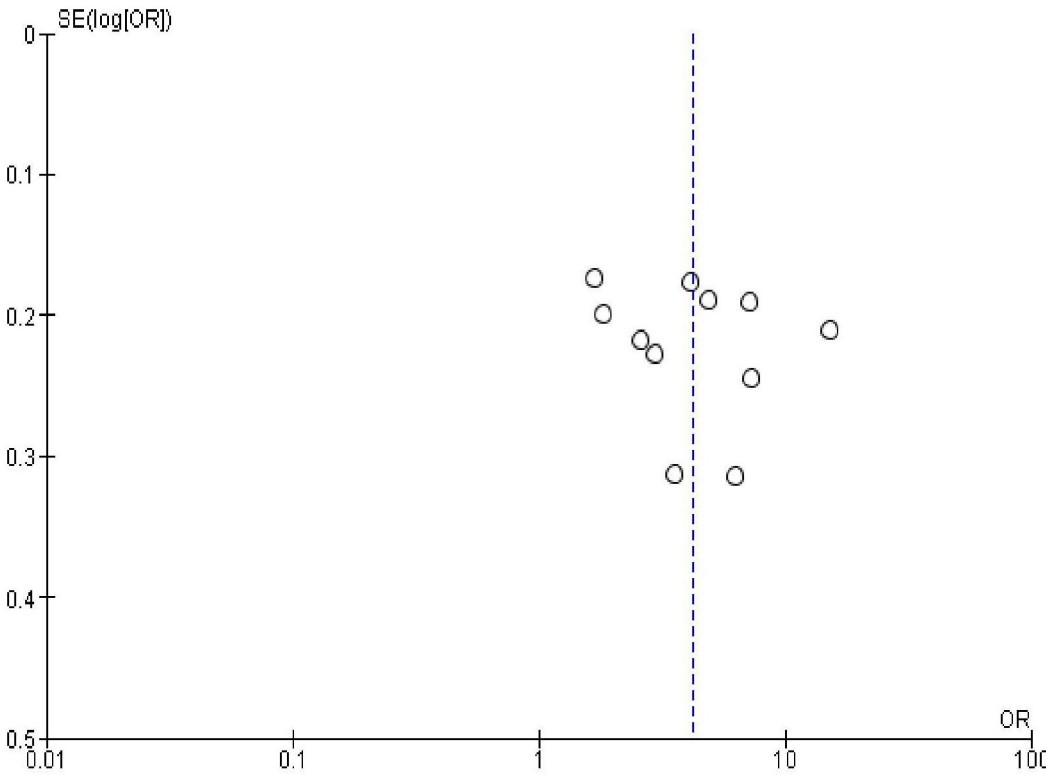

**Fig 22. Publication bias on having PNC follow up.**

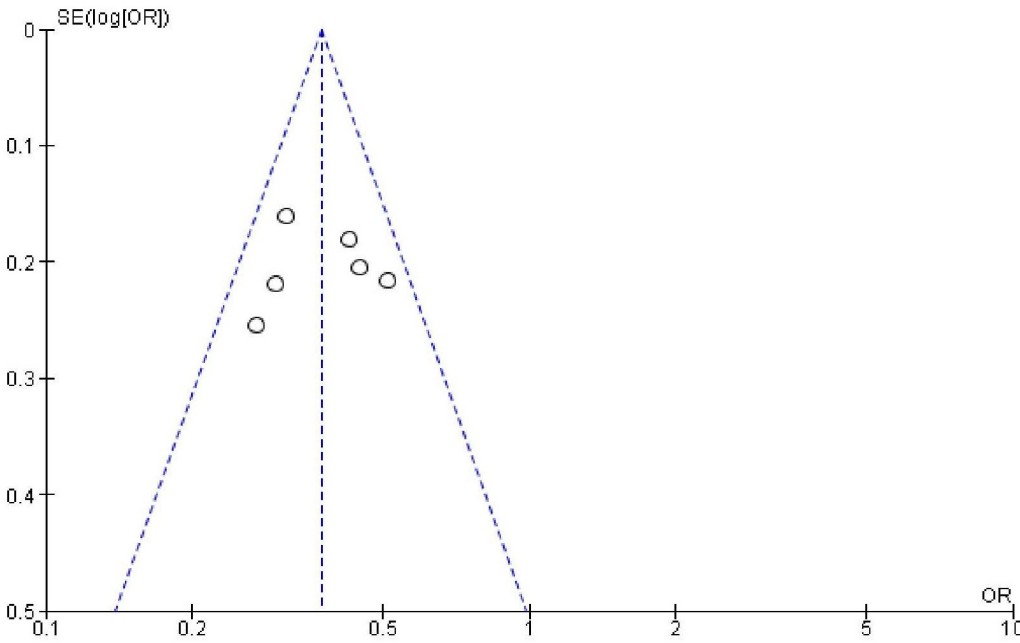

**Fig 23. Publication bias on length of time after delivery.**

health system money, and responds to the multiple demands on women's time that often leads them to neglect their own health [39].

Sexual and reproductive characteristics related factors are resumption of sexual intercourse after birth [OR = 6.22; 95% CI: 3.01, 12.86], menses return after birth [OR = 3.72; 95% CI: 1.98, 6.99], discussion about family planning with partner [OR = 2.53; 95% CI: 2.00, 3.20] and ever heard about modern contraceptives use during postpartum period [OR = 4.93; 95% CI: 2.26, 10.76]. From this group of factors resumption of sexual intercourse after birth, ever heard about modern contraceptives use during postpartum period and menses return after birth are the stronger predictors respectively. This may imply the need for providing contraceptive messages to women and their partner focusing on pregnancy risks even prior to menses return as the unprotected sexual intercourse during early postpartum period can put them at higher risk to pregnancy. Such educational interventions help increase knowledge of available contraceptive methods, enabling women to make informed decisions and use contraception more effectively [40]. Hence, sustainable community based behavioral change communications (BCC) to women could be used as an intervention to improve contraceptives uptake in the area.

In our review, women's educational level and marital status are the socio-demographic characteristics that showed association with modern contraceptive uptake in the first year after birth. Women who had formal education were 2.6 times more likely to utilize modern contraceptives during postpartum period compared to women who did not have formal education. This finding is similar with a previously conducted systematic review and meta-analysis study in Ethiopia [11]. This might be due to the fact that the more educated women have better understanding of benefits of contraceptives during postpartum period and risks of shorter inter-pregnancy interval. As a result, they may visit health facilities and uptake it. Therefore, concerned stakeholders should work in improving women's education as this could improve contraceptives uptake during postpartum period.

A married woman was 4.3 times more likely to initiate modern contraceptives during postpartum period compared to unmarried women. Possible explanation of this finding could be that a married woman may be encouraged and supported by her partner to use contraceptive during postpartum period. Our finding is in line with a systematic review and meta-analysis on postpartum contraceptive use among women in low- and middle-income countries [41]. The compared study reported that married women were more likely to use contraception than single women, as were women who reported they had support from their partner to use contraceptives. The same study also reported lower contraceptive use among women without current partners, who may have less need for contraception due to lack of, or infrequent, sexual activity. But, our finding is not consistent with the study conducted in Sub-Saharan Africa, Latin America, and the Caribbean countries [42]. In the mentioned study, unmarried women tend to have a higher prevalence of contraceptive use compared to their married counterparts as unmarried women usually want to avoid becoming pregnant. Thus, despite more infrequent sexual activity compared with married women, they also are more likely than their married counterparts to negotiate contraceptive use with their partners. The possible reason for the inconsistency is that our review was done on contraceptives use during the first year after birth but the compared study was conducted on an extended period. Hence, our finding may imply the need for paying more attention to women who are not married.

This review was not free from limitations. Most of the included studies are cross-sectional which limits the ability to assess cause-effect relationship between contraceptive uptake and the considered factors. We also noticed considerable heterogeneity among the studies, which could be due to the differences in the study area, period, and other factors. In our meta-analysis, we wanted to do subgroup analysis by regions because one factor may be more important in one region and other factors in other regions. Unfortunately we couldn't do that because there are

only 1 or 2 studies from regions. We suggest future researches on PPFP should consider these limitations. Despite these limitations, the review had reportable strengths. Studies were included from all the bigger regions in Ethiopia Oromia, Amhara, SNNP, Addis Ababa & Tigray), which are home to more than 85% of Ethiopian population. Methodologically, the quality of the included studies was generally acceptable. They used adequate sample size, subjects were randomly selected, and used validated tools. The methodology of the systematic review and meta-analysis was rigorous. Extensive and comprehensive search strategies were used to access as many studies as possible from multiple databases. Studies were also evaluated for methodological quality using recommended standardized tools. The risk of bias of the included studies was also found to be low. In our review we tried to show factors associated in theme: socio-demographic, sexual and reproductive health activities and services. We also tried to identify the stronger predictors from each group of factors to indicate to which issues stakeholders should strongly work to improve the contraceptive uptake rate during the 1st year after birth. We belief this can add.

## Conclusions and recommendations

In our review, we identified a number of predictors for modern contraceptive uptake during the first year after birth. An interesting issue is that all these predictors are modifiable and most of them are related to reproductive health characteristics and services. These may imply the need for integrating postpartum family planning into reproductive, maternal, new-born and child health services particularly at primary health care system and community level to improve contraceptive uptake during postpartum period in Ethiopia. Postpartum women who are eligible for PPFP should be counseled at all RMNCH service delivery points and referred to Family planning clinic/room so that they can receive the method of their choice. We believe this can reduce the high unmet need for contraceptives. Hence, we suggest that key stakeholders like Ministry of health, Regional health bureaus, health managers, health care providers, health extension workers, and other factors should strongly work on the integration of these services. We also suggest providing contraceptive messages to women and their partner focusing on pregnancy risks even prior to menses return as the unprotected sexual intercourse during early postpartum period can put them at higher risk to pregnancy.

## Supporting information

**S1 File. Prospero registry.**
(PDF)

**S2 File. PRISMA 2009 checklist.**
(PDF)

**S1 Table. Search method used.**
(DOCX)

**S2 Table. Data extraction template.**
(XLSX)

**S3 Table. Data extracted from the included studies.**
(DOCX)

## Acknowledgments

We would like to thank Arsi University for allowing us to use the internet to search the electronic databases.

## Author Contributions

**Conceptualization:** Gebi Husein Jima, R. G. Biesma-Blanco, Tegbar Yigzaw Sendekie, J. Stekelenburg.

**Data curation:** Gebi Husein Jima, Muhammedawel Kaso Kaso.

**Formal analysis:** Gebi Husein Jima.

**Investigation:** Gebi Husein Jima.

**Methodology:** Gebi Husein Jima, Muhammedawel Kaso Kaso, R. G. Biesma-Blanco, Tegbar Yigzaw Sendekie, J. Stekelenburg.

**Resources:** Gebi Husein Jima.

**Software:** Gebi Husein Jima.

**Supervision:** R. G. Biesma-Blanco, Tegbar Yigzaw Sendekie, J. Stekelenburg.

**Validation:** Gebi Husein Jima, R. G. Biesma-Blanco, Tegbar Yigzaw Sendekie, J. Stekelenburg.

**Visualization:** Gebi Husein Jima, R. G. Biesma-Blanco, J. Stekelenburg.

**Writing – original draft:** Gebi Husein Jima.

**Writing – review & editing:** Gebi Husein Jima, Muhammedawel Kaso Kaso, R. G. Biesma-Blanco, Tegbar Yigzaw Sendekie, J. Stekelenburg.

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
