## [Decision Letter · Decision Letter 0]

14 Apr 2022

PONE-D-22-00169Factors associated with modern contraceptives uptake during the first year after birth in Ethiopia: a systematic review and meta-analysisPLOS ONE

Dear Dr. Jima,

Thank you for submitting your manuscript to PLOS ONE. After careful consideration, we feel that it has merit but does not fully meet PLOS ONE’s publication criteria as it currently stands. Therefore, we invite you to submit a revised version of the manuscript that addresses the points raised during the review process. Please submit your revised manuscript by May 29 2022 11:59PM. If you will need more time than this to complete your revisions, please reply to this message or contact the journal office at plosone@plos.org. Please include the following items when submitting your revised manuscript:A rebuttal letter that responds to each point raised by the academic editor and reviewer(s). You should upload this letter as a separate file labeled 'Response to Reviewers'.A marked-up copy of your manuscript that highlights changes made to the original version. You should upload this as a separate file labeled 'Revised Manuscript with Track Changes'.An unmarked version of your revised paper without tracked changes. You should upload this as a separate file labeled 'Manuscript'.

We look forward to receiving your revised manuscript.

Kind regards,

Dylan A Mordaunt

Academic Editor

PLOS ONE

Journal Requirements:

2. We note that you have referenced (ie. Bewick et al. [5]) which has currently not yet been accepted for publication. Please remove this from your References and amend this to state in the body of your manuscript: (ie “Bewick et al. [Unpublished]”) as detailed online in our guide for authors

Additional Editor Comments (if provided):

Thank you for your submission. The reviewers point to a number of major factors that would need to be addressed. In particular they've pointed to the replicative nature of the study. PLoS One' critieria for publication do not require novelty but rather originality- this is why I've recommended major revision rather than rejection. In saying that, a repeat of a systematic review that covers an area previously addressed in the literature, would need a good reason for us to accept this such as that there were new findings, that the scope (i.e. PICOT) is different or some other reason. I will leave it to the authors to explain how your study is original.

1. The study may represent the results of original research, though this is unclear currently.

2. Results do not appear to have been reported elsewhere.

3. Experiments, statistics, and other analyses have significant issues outlined by the reviewers.

4. Conclusions are presented in an appropriate fashion and are supported by the data.

5. The article is presented in an intelligible fashion and is written in standard English.

6. The research meets all applicable standards for the ethics of experimentation and research integrity.

7. The article should adhere to relevant reporting standards such as PRIMSA with the checklist or diagram included. I would also suggest review with AMSTAR 2 and inclusion of the resulting completed score.

Reviewers' comments:

Reviewer's Responses to Questions

**Comments to the Author**

1. Is the manuscript technically sound, and do the data support the conclusions?

Reviewer #1: Partly

Reviewer #2: Yes

2. Has the statistical analysis been performed appropriately and rigorously? 

Reviewer #1: No

Reviewer #2: Yes

3. Have the authors made all data underlying the findings in their manuscript fully available?

Reviewer #1: Yes

Reviewer #2: Yes

4. Is the manuscript presented in an intelligible fashion and written in standard English?

Reviewer #1: Yes

Reviewer #2: Yes

5. Review Comments to the Author

Reviewer #1: Jima and colleagues present a manuscript that investigates the determinants of postpartum family planning in Ethiopia by using systematic review and meta-analysis. Though the analysis is weak, the manuscript does not contribute any new information on the determinants of postpartum family planning that we don’t already know. There are been many previous systematic review and meta-analysis articles that analyzed postpartum family planning in Ethiopia, so the Authors need to describe the novelty of the research. It is good for the authors to find the determinants of postpartum family planning that specific based on the cultural or specific condition in Ethiopia. For the funnel plots of postpartum family planning in Ethiopia, the authors should include all the studies used in the analysis.

Reviewer #2: 1. The authors mention that they included all regions of Ethiopia in their paper. It would be better to mention what are the special differences in those regions that are missed in previous systematic review by Mehare et al and included in the current review.

2. I do not clearly see newer findings from the current review compared to previous review papers from Ethiopia. Authors need to highlight if they found new findings by including new regions.

3. Principally, I do not suggest conducting a metanalysis of such studies where differences in research contexts i.e cultural factors, supply side barriers and so many other aspects affect FP utilization. Authors should provide the objective of conducting metanalysis and see if conducting a metanalysis fulfills their objectives. Maternal education could be more important in one region whereas household wealth may be more important in other regions. Implementing a blanket approach based on findings from such metanalysis will not improve post-partum family planning.

4. Authors have reported that use of ANC, institutional births and other services are associated with higher uptake of FP. On the other hand, they pool data from community-based studies and facility-based studies together. It would be more useful if they only selected community based studies. or studied community-based studies and facility based studies separately.

5. I suggest only limit to a systematic review of factors and study factors separately from community based studies and facility based studies.

6. PLOS authors have the option to publish the peer review history of their article (what does this mean?). If published, this will include your full peer review and any attached files.

Reviewer #1: No

Reviewer #2: No

---

## [Author Response · Author response to Decision Letter 0]

27 May 2022

To: PLOS ONE journal editorial office

Re: PONE-D-22-00169

 Factors associated with modern contraceptives uptake during the first year 

 after birth in Ethiopia: a systematic review and meta-analysis

Author: Gebi Husein Jima et al.

Dear Academic editor,

First of all, we would like to thank you for considering our manuscript for publication in this top ranked journal. We also want to thank the peer reviewers for giving us constructive feedback and comments to improve our work. We have carefully considered all your and reviewers’ comments and improved our manuscript. We also revised our manuscripts to meet PLOS ONE's style requirements, including issues related to headings, figures & tables citations and captions, references sections and file naming. We have put track changes in the manuscript to highlight changes we made: Its file name is 'Revised Manuscript with Track Changes'. We also separately worked an unmarked version of the revised manuscript with a file name ‘Manuscript’. Moreover, we have described below the point-by-point responses to your and the peer reviewers’ comments. 

A. Point by point responses to comments provided by Academic editor:

Journal Requirements:

Authors’ response: Dear Editor, Thank you for your valued comment. We revised our manuscript after carefully considering PLOS ONE's style requirements for issues related to headings, figures’ & tables’ citations and captions, reference sections and files naming. 

2. We note that you have referenced (ie. Bewick et al. [5]) which has currently not yet been accepted for publication. Please remove this from your References and amend this to state in the body of your manuscript: (ie “Bewick et al. [Unpublished]”) as detailed online in our guide for authors

Authors’ response: Thank you Editor again for your comment. We removed reference number 37 from reference list. We also changed its citation to (Belete GA et al, 2019[Unpublished]). See table 1, row # 22 on page # 7 of the revised manuscript.

Additional Editor Comments (if provided):

Thank you for your submission. The reviewers point to a number of major factors that would need to be addressed. In particular they've pointed to the replicative nature of the study. PLoS One' critieria for publication do not require novelty but rather originality- this is why I've recommended major revision rather than rejection. In saying that, a repeat of a systematic review that covers an area previously addressed in the literature, would need a good reason for us to accept this such as that there were new findings, that the scope (i.e. PICOT) is different or some other reason. I will leave it to the authors to explain how your study is original.

1. The study may represent the results of original research, though this is unclear currently.

2. Results do not appear to have been reported elsewhere.

3. Experiments, statistics, and other analyses have significant issues outlined by the reviewers.

4. Conclusions are presented in an appropriate fashion and are supported by the data.

5. The article is presented in an intelligible fashion and is written in standard English.

6. The research meets all applicable standards for the ethics of experimentation and research integrity.

7. The article should adhere to relevant reporting standards such as PRIMSA with the checklist or diagram included. I would also suggest review with AMSTAR 2 and inclusion of the resulting completed score.

Authors’ response: Thank you Editor for considering our work for publication in you top rank journal (PLOS ONE). We also thank the peer reviewers for pointing major issues in our manuscript that help us improve our work. We particularly considered your concern about a repeat of a systematic review that covers an area previously addressed. Based on your suggestion, we addressed this issue under the comments of reviewers below (see page # 3-5 of this rebuttal letter). 

With regards to the use of relevant reporting standards, we already used PRISMA diagram (see fig 1) to show the overall study selection process. We also used PRISMA Checklist (see ‘S2_File.pdf’ which is supplemental file). But with your suggestions to use AMSTAR 2, the checklist is intended for randomized and/or non-randomized studies of healthcare interventions. In our case, studies included are not interventional studies; they are observational studies (cross-sectional, case control and cohort studies). So we don’t think this checklist is relevant to our work.

B. Point by point responses to comments provided by reviewer # 1:

Reviewer #1: Jima and colleagues present a manuscript that investigates the determinants of postpartum family planning in Ethiopia by using systematic review and meta-analysis. Though the analysis is weak, the manuscript does not contribute any new information on the determinants of postpartum family planning that we don’t already know. There are been many previous systematic review and meta-analysis articles that analyzed postpartum family planning in Ethiopia, so the Authors need to describe the novelty of the research. It is good for the authors to find the determinants of postpartum family planning that specific based on the cultural or specific condition in Ethiopia. For the funnel plots of postpartum family planning in Ethiopia, the authors should include all the studies used in the analysis.

Authors’ response: We thank reviewer for the constructive comment. As you said, there are previously done systematic review and meta-analysis studies that addressed postpartum family planning in Ethiopia. At the time of our review, there were two systematic review and meta-analysis studies done on postpartum family planning uptake in Ethiopia. These are: Wakuma B, Mosisa G, Etafa W, Mulisa D, Tolossa T, Fetensa G, et al. Postpartum modern contraception utilization and its determinants in Ethiopia: A systematic review and meta-analysis. PLS ONE. 202; 15(12): e0243776. https://doi.org/10.1371/journal.pone.0243776 and Mehare T, Mekuriaw B, Belayneh Z, Sharew Y. Postpartum Contraceptive Use and Its Determinants in Ethiopia: A systematic Review and Meta-anamysis. International Journal of Reproductive Medicine. 2020; 5174656. https://doi.org/10.1155/2020/5174656 . We carefully reviewed these two articles and identified the gaps which we should address in our review. One of the gaps we saw in these articles is the country representativeness. Most of the included articles are from few regions. Oromia which is the largest region and home for 40% of the total population of the country is not represented in both reviews. In Ethiopia there is a significant difference between regions with regards to maternal and child health service utilization including contraceptives. You can see Ethiopia Demographic and Health Survey 2019 which is the latest DHS in Ethiopia(Citation: Ethiopian Public Health Institute (EPHI) [Ethiopia], ICF. Ethiopia Mini Demographic and Health Survey 2019: Key Indicators. Rockville, Maryland, USA: EPHI and ICF, 2019). That is why we always consider regions especially in studies related to maternal and child health services use. In Wakuma B, et al: from 19 studies included in to the review, 50% are from a single region (Amhara) and 63% is only from the Northern part of the country( Amhara & Tigray). Only 2 are from Oromia. In Mehare Tet al: from the total of 18 primary studies included in the review, 61% are only from the Northern part of Ethiopia (Amhara & Tigray) and only one study is from Oromia. For your information, data from Oromia region is representative of the national data. For this strong reason, we don’t think the included articles in both reviews can represent the situation in Ethiopia. In our review, the major regions are adequately represented (Oromia, Amhara, Addis Ababa and SNNP).

In addition, in the mentioned two previously done systematic review and meta-analysis, ‘Antenatal care follow-up’ alone was statistically associated with family planning uptake. Whereas in our case, ‘ANC follow-up where family planning was discussed’ had shown association with contraceptive uptake. This can urge for counseling about contraceptive during ANC so that women can uptake it immediately after birth. So, we feel this is one important finding in our work. Another important finding in our review was that ‘Facility birth’ was found to be a predictor for contraceptive uptake during postpartum period. This was not reported in the previously done systematic review and meta-analysis studies in Ethiopia. This implies when women are given information about contraceptives as soon as possible after birth, then they can uptake it within the postpartum period and delay the next pregnancy. The remaining predictors in our review are similar with the predictors reported in the previous reviews which show the consistency/replicable nature of the findings. The intention of our review is not only to come up with new factors; it is to see consistencies of finding with the previously reported results. 

Dear reviewer, we also agree with your suggestion to identify the determinants of postpartum family planning that are specific to cultural or condition in Ethiopia. In Ethiopia Maternal and child health services utilization is highly influenced by the community’s culture and belief which could be further influenced by other specific conditions. That is the reason why women’s socio-demographic characteristics like their marital status and education were considered in our review. In addition, reproductive health services related factors: like women’s contact with the health system for the basic services, PPFP counseling during these basic services, facility birth can affect contraceptives uptake during postpartum period. Moreover, sexual & reproductive related factors: like menses return after birth, resumption of sexual intercourse after birth, discussion about family planning with partner, previous information about contraceptives use during postpartum period, having experience of contraceptive use before are also influential. It is for this reason we grouped and discussed predictors in to socio-demographic, reproductive health services and sexual & reproductive related factors (see discussion section on page # 11-13 of the revised manuscript).

For the suggestion to include all funnel plots, we worked funnel plot for all variables and uploaded. See ‘fig 13-23’. Manuscript is also revised (see page # 11 of the revised manuscript).

C. Point by point responses to comments provided by reviewer # 2:

1. The authors mention that they included all regions of Ethiopia in their paper. It would be better to mention what are the special differences in those regions that are missed in previous systematic review by Mehare et al and included in the current review.

Authors’ response: Dear reviewer, thank you for the constructive comment. At the time of our review, there were two systematic review and meta-analysis studies done on postpartum family planning uptake in Ethiopia. One of these two is the one that you mentioned ( Mehare et al). We carefully reviewed these two articles and identified the gaps which we should address in our review. One of the major gaps we observed in ‘Mehari et al’ and of course in the other article is representativeness issue. Most of the included articles are from few regions. 61% of the included studies are only from the Northern part of Ethiopia (Amhara and Tigray). There is only one study from Oromia region which is the largest region in Ethiopia and home for 40% of the total population of the country. For your information, data from Oromia region is representative of the national data. In our review, the major regions are adequately represented (Oromia, Amhara, Addis Ababa and SNNP).We always consider this because in Ethiopia there is a significant difference between regions with regards to maternal and child health service utilization including contraceptives. You can see Ethiopia Demographic and Health Survey 2019 which is the latest DHS in Ethiopia(Citation: Ethiopian Public Health Institute (EPHI) [Ethiopia], ICF. Ethiopia Mini Demographic and Health Survey 2019: Key Indicators. Rockville, Maryland, USA: EPHI and ICF,2019). That is why we always consider regions in studies related to health services use. 

2. I do not clearly see newer findings from the current review compared to previous review papers from Ethiopia. Authors need to highlight if they found new findings by including new regions.

Authors’ response: Dear Reviewer, thank you for your concern. As we mentioned above, during our review, there were only two systematic review and meta-analysis studies done on postpartum contraceptive uptake in Ethiopia. In these two systematic review and meta-analysis, ‘Antenatal care (ANC) follow-up’ alone was statistically associated with family planning uptake after birth. But in our study that was not true. In our case, ‘ANC follow-up where family planning was discussed’ had shown association with contraceptive uptake after birth. This can urge for counseling about contraceptive during ANC so that women can uptake it immediately after birth. So, we feel this is one important finding in our work. Another important finding in our review was that ‘Facility birth’ was found to be a predictor for contraceptive uptake during postpartum period. This was not reported in the previously done systematic review and meta-analysis studies in Ethiopia. This implies when women are given information about contraceptive as soon as possible after birth, then they uptake it with in postpartum period to delay the next pregnancy.

3. Principally, I do not suggest conducting a metanalysis of such studies where differences in research contexts i.e cultural factors, supply side barriers and so many other aspects affect FP utilization. Authors should provide the objective of conducting metanalysis and see if conducting a metanalysis fulfills their objectives. Maternal education could be more important in one region whereas household wealth may be more important in other regions. Implementing a blanket approach based on findings from such metanalysis will not improve post-partum family planning.

Authors’ response: Thank you reviewer for your comment again. We would like to ensure you that we have a clear objective for our systematic review and meta-analysis. That is ‘to identify and summarize predictors of postpartum family planning uptake during postpartum period in Ethiopia’. And we feel that we have addressed this objective through our review. On the other hand, as you said one factor may be more important in one region and the other factors in the other region. We quite agree with your point. To address this issue, we tried to do subgroup analysis by regions. But the problem was, there are only few studies (1 or 2) from even major regions and no studies from smaller regions. For this reason, we couldn’t do subgroup analysis for even larger regions. Considering your suggestion, we include this in our revised manuscript as a limitation of our review so that the researcher in the future can address it (see page # 13, line 12-15 of the revised manuscript).

4. Authors have reported that use of ANC, institutional births and other services are associated with higher uptake of FP. On the other hand, they pool data from community-based studies and facility-based studies together. It would be more useful if they only selected community based studies. or studied community-based studies and facility based studies separately.

Authors’ response: We thank the reviewer for the comment. Frist, ‘ANC follow up’ alone is not predictor in our study. It is ‘ANC follow up where contraceptive was discussed’. On the other hand, as you said, we didn’t separately analyze community based and facility based studies. In our review, we included 16 community based studies and 6 facility based studies. Even the 6 facility based studies included were based on Primary health care units (Health centers) not from bigger health facilities/Hospitals. In Ethiopia, 85% of the people reside in rural area and small towns. Primary health care units are located at district level where it is accessible to these majority people unlike hospitals. So the 6 facility based studies have no major difference from the remaining community based studies. That is why we included all. 

5. I suggest only limit to a systematic review of factors and study factors separately from community based studies and facility based studies.

Authors’ response: We thank the reviewer for the comment. In our review, we included 16 community based studies and 6 facility based studies. Even the 6 facility based studies were based on Primary health care units (PHCUs) not from bigger health facilities/Hospitals. In Ethiopia, 85% of the people reside in rural area and small towns. Primary health care units are located at district level where it is accessible for these majority people unlike hospitals. So the 6 facility based studies have no major difference from the remaining community based studies. That is why we didn’t separate facility based and community based in our analysis.

6. PLOS authors have the option to publish the peer review history of their article (what does this mean?). If published, this will include your full peer review and any attached files.

Do you want your identity to be public for this peer review? For information about this choice, including consent withdrawal, please see our Privacy Policy.

Authors’ response: yes

Authors’ response: there are no attachment files. "View Attachments". If this link does not appear

Authors’ response: All figures are uploaded to PACE and converted to “tif” based on your comment.

Best regards,

Gebi Husein Jima on behalf the authors

---

## [Editor Report · Decision Letter 1]

3 Jun 2022

Factors associated with modern contraceptives uptake during the first year after birth in Ethiopia: a systematic review and meta-analysis

PONE-D-22-00169R1

Dear Dr. Jima,

We’re pleased to inform you that your manuscript has been judged scientifically suitable for publication and will be formally accepted for publication once it meets all outstanding technical requirements.

Kind regards,

Dylan A Mordaunt, MD, MPH, FRACP

Academic Editor

PLOS ONE

Additional Editor Comments (optional):

Thank you for your resubmission. This now meets the criteria for publication.
---

## [Editor Report · Acceptance letter]

7 Jul 2022

PONE-D-22-00169R1 

actors associated with modern contraceptives uptake during the first year after birth in Ethiopia: a systematic review and meta-analysis 

Dear Dr. Jima:

I'm pleased to inform you that your manuscript has been deemed suitable for publication in PLOS ONE. Congratulations! Your manuscript is now with our production department. 

Kind regards, 

on behalf of

Associate Professor Dylan A Mordaunt 

Academic Editor

PLOS ONE